# Atomic Research on the Diffusion Behavior, Mechanical Properties and Fracture Mechanism of Fe/Cu Solid–Liquid Interface

**Hongyu Zheng \*, Jingwen Sun, Na Guo and Mingjie Wang \***

School of Intelligent Manufacturing, Huanghuai University, Zhumadian 463000, China
\* Correspondence: 20191887@huanghuai.edu.cn (H.Z.); 15513882577@163.com (M.W.)

**Abstract:** A molecular dynamics simulation was applied to investigate the diffusion behavior and mechanical properties of a Fe/Cu solid–liquid interface with different orientations, temperatures, and strain rates. The results show that the displacement distance of Fe atoms' diffusion into the Cu matrix was obviously larger than that of Cu atoms' diffusion into the Fe matrix at any diffusion temperature and diffusion time. Moreover, the diffusion coefficient and diffusion distance both increase with temperature and time, and reach the highest value when the temperature and diffusion time are 1523 K and 3 ns, respectively. Additionally, the diffusion coefficients of the Fe atoms are arranged in the following order: Fe (100) < Fe (110) < Fe (111). The diffusion coefficients of the Cu atoms are arranged in the following order: Cu (110) > Cu (111) > Cu (100), when temperature and time are 1523 K and 3 ns, respectively. The yield strength and fracture strain of the bimetallic interface is positively correlated with the strain rate, but negatively correlated with the tensile temperature. Moreover, the yield strength of the three orientations can be arranged as follows: Fe (110)/Cu (110) > Fe (100)/Cu (100) > Fe (111)/Cu (111), and the yield strength and fracture strain of Fe (110)/Cu (110) diffusion interface are 12.1 GPa and 21% when the strain rate was $1 \times 10^9$/s and the tensile temperature was 300 K. The number of stacking faults and dislocations of the diffused Fe/Cu interface decreased significantly in comparison to the undiffused Fe/Cu interface, even in the length of Stair-rod dislocation and Shockley dislocation. All these results lead to a decrease in the tensile yield strength after interface diffusion.

**Keywords:** diffusion behavior and mechanical properties; Fe/Cu solid–liquid interface; molecular dynamics; orientation; temperature and strain rate



## 1. Introduction

Over the last few years, high-strength and highly wear resistant bimetallic materials have been attracted worldwide attention and demonstrated important application prospects in aerospace, ships, automobiles, and other fields due to their excellent dual performance structure and the comprehensive high-strength and high wear resistance properties [1–5]. The steel–copper bimetallic materials are one of the typical representatives, combining the advantages of both steel (high strength and stiffness) and copper (high wear resistance), and have been used to fabricate military aerospace rotors [6]. Additionally, for steel–copper bimetallic materials with a high bonding strength, the relationship between the diffusion mechanism, the diffusion distance, and the tensile stress-strain of the interface atoms play a more important role in determining the bonding strength and the diffusion coefficient of the bimetallic interface [7,8]. However, the pivotal technique of steel–copper bimetallic materials, the interfacial diffusion and interface enhancement mechanism of steel–copper bimetallic materials, has hardly been published and investigated.

Up to now, atomic calculations have been demonstrated to be an effective method to comprehensibly investigate the solid–liquid interface, as it is difficult to carry out an experimental investigation at high temperature [9–11]. According to the relevant reports,

several theoretical investigations of solid–liquid interfaces and multi-layer interfaces have been proposed, such as the Cu/Pb interface [12,13], Al/Cu interface [14,15], Ni/Al multi-layer [16], Al/Al$_2$O$_3$ interface [17], and Fe/Li interface [18]. Especially, Mao et al. [19] studied the diffusion behavior of the Cu/Al solid–liquid interface using the molecular dynamic (MD) method, and found that the influence of temperature and pressure on the diffusion behavior of Cu/Al solid–liquid interface is greater than that of time, and all of these results depend on the formation of θ−Al$_2$Cu. S. Raman et al. [20] investigated the thermodynamic and kinetic properties of Fe/Mn solid–liquid interface utilizing MD simulations, and successfully revealed that the average value of solid–liquid interfacial free energy and anisotropy parameters remain constant with fluctuating temperature and the variation in diffusion speed is higher than that of the bulk liquid diffusion coefficient with increasing temperature. Moreover, a number of researchers have explored some incurable factors associated with the mechanical properties of solid–liquid interface, such as stress–strain curves and microstructure evolution in the course of the tensile process [21–23]. Su et al. [24] investigated the tensile behavior of Ti/Ni multilayered films and predicted the tensile process, stress–strain curve, and other tensile behaviors through MD simulations. Liu et al. [25] simulated the diffusion process of the Al/Cu interface, analyzed the tensile behavior, and improved the mechanical properties of the Al/Cu solid–liquid interface after diffusion solidification and cooling. All these mechanical property investigations of the solid–liquid interface have been studied by quite a few scholars, but there has been no systematic analysis of the Fe/Cu solid–liquid interface.

In this paper, we aimed to propose a theoretical framework based on the diffusion behavior and mechanical properties of Fe/Cu solid–liquid interface using MD simulation. In detail, we analyzed the diffusion coefficient, the diffusion distance, the atomic concentration distribution, and the diffusion mechanism of Fe/Cu solid–liquid interfaces at different temperatures and diffusion times. Meanwhile, the mechanical properties, i.e., tensile strain–stress curve, influence of tensile orientation, strain rate, and tensile temperature of the Fe/Cu solid–liquid interface, are also investigated. Although the diffusion and bonding characteristics of steel–copper bimetallic interfaces cannot be directly characterized in time scale and space scale in this study, our calculations provide a theoretical guidance for the production and manufacture of high-performance Fe/Cu bimetallic materials.

## 2. Simulation Methodology

All MD simulations are calculated using the Large-scale Atomic/Molecular Massively Parallel Simulator (LAMMPS) [26] with a time step of 1.0 fs. Additionally, an embedded atomic method (EAM) potential developed by Byeong-Joo Lee et al. [27] was used for modeling the atomic interaction between FCC Fe atoms and FCC Cu atoms near the interface, which has been employed in exploring diffusion properties, mechanical properties [28], and magnetic properties [29–31] of the Fe–Cu alloys. An open visualization tool (OVITO) [32] was used to observe diffusion behavior and the deformation process. Moreover, periodic boundary conditions were applied in three transverse (x, y, and z) directions. The initial velocity of atoms was assumed to accord with Maxwell–Boltzmann random distribution. Additionally, the Verlet integration algorithm was introduced to solve the Newtonian equation of motion integration.

In this study, three surface orientations ((100), (110), and (111)) of the γ-Fe/Cu interface were used to calculate the diffusion behavior and mechanical properties of Fe/Cu bimetallic materials. An additional advantage of using a case study approach is that our simulation models are based on our steel–copper bimetallic casting experiment at high temperature [33,34]. The modeling consists of FCC Fe bulk and FCC Cu bulk, and the initialization configurations of a cross-section are shown in Figure 1. As seen in Figure 1, the initial Fe/Cu interface model consisted of two parts: an upper Cu block containing 6400 atoms with the size of $10a_{Cu} \times 5a_{Cu} \times 30a_{Cu}$ and a lower Fe block with the same number of atoms and the size of $10a_{Fe} \times 5a_{Fe} \times 30a_{Fe}$ (the lattice constant of the $a_{Cu}$ and $a_{Fe}$ atoms are 0.36147 and 0.36457 nm respectively). To perform the solid–liquid interface

calculations, five different temperatures—1323, 1373, 1423, 1473, and 1523 K—were selected according to the melting point of Cu and Fe. Additionally, the overall diffusion process was relaxed at 300 K under a constant pressure-temperature (NPT) ensemble for 100 ps, and then relaxed at the target diffusion temperature under a constant pressure-temperature (NPT) ensemble for 1000 ps. Finally, the system was quickly cooled to 300 K, shown in Figure 1.

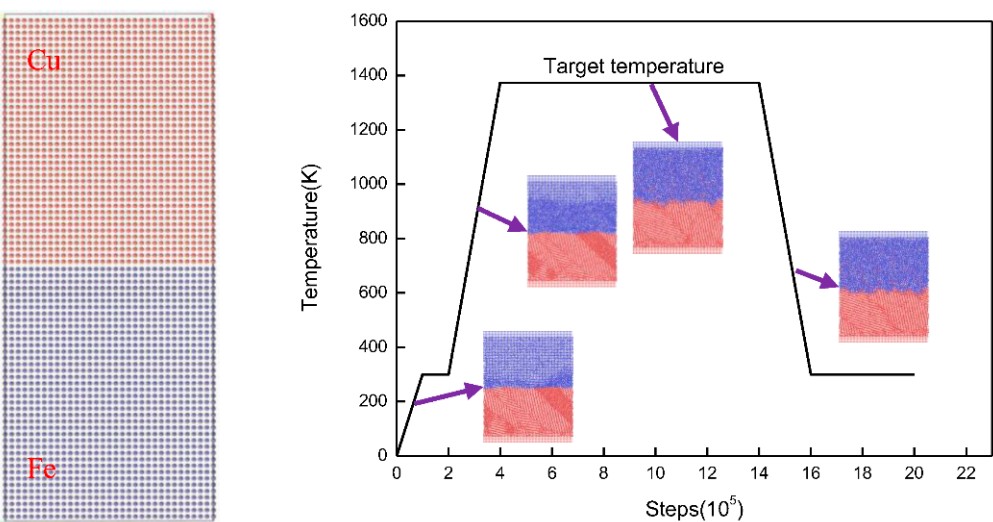

**Figure 1.** Atomic configuration of Fe/Cu interface model and the temperature–time relationship of the diffusion process.

According to [35], the mean square displacement (MSD) of the solid–liquid interface can be approximately calculated in such a complicated system. Thus, the MSD of Cu and Fe was calculated first to forecast the diffusion coefficient. The MSD can be defined by the following equation [36]:

$$\text{MSD} = \left\langle r^2(t) \right\rangle = \left\langle |r_i(t) - r_i(0)|^2 \right\rangle$$

where $r_i(t)$ is the position vector of atom $i$ at time $t$, which represents the average ensemble of the atoms in the simulated time. Accordingly, the diffusion coefficient is defined by the following equation [37]:

$$\text{D} = \lim_{t \to \infty} \frac{1}{2Nt} \left\langle |r(t) - r(0)|^2 \right\rangle$$

where D is the diffusion coefficient, $N$ denotes the dimension of the simulated system, and $N = 3$ for the simulation of blocks.

## 3. Result and Discussion

### 3.1. Diffusion Behavior

Temperature and diffusion time, as the two main factors determining the Fe/Cu interface, were considered first in our simulation. Thus, the snapshots of the cross-section diffusion process of Fe bulk and Cu bulk at different temperatures and times are distinctively illustrated in Figure 2. By comparing the four snapshots under the same temperature with the diffusion time elevated, the portion of atoms diffusing across the initial Fe/Cu interface can be seen to increase gradually and the diffusion depth of the Fe atoms diffusing into the Cu bulk is deeper, while that of Cu atoms diffusing into the Fe bulk is shorter. From these snapshots, it can be seen that only a small amount of Fe atoms cross through the initial interface, while much less Cu atoms cross the interface at 0.5 ns. After 3.0 ns, the diffusion distance of Cu atoms and Fe atoms both increase, the percentage of Fe atoms that diffuse across the interface increases to 1.5% from 0.3% at 0.5 ns, and the percentage of Cu atoms increases to 0.3% from 0.05% at 0.5 ns. These asymmetrical diffusion phenomena

have similarly appeared in Al/Cu [38], Mo/Ti [39], and Fe/W [40] interfaces. Likewise, by comparing five different temperatures at the same diffusion time, one can see that only a few Fe atoms diffuse into the Cu bulk at 1323 K, the interfacial structure is relatively smooth, and the degree of atomic confusion is lower. However, when temperature is increased, the portion of Fe and Cu atoms that cross initial interface increases and the diffusion distance of Fe atoms diffusing into the Cu bulk is also deeper than that of the Cu atoms diffusing into the Fe bulk at every temperature. Meanwhile, when the temperature is increased to 1523 K from 1323 K with a 50 K temperature interval, the percentages of the number of Fe atoms diffusing into the Cu bulk are 0.1%, 0.13%, 0.15%, 0.17%, and 0.2%, while that of the Cu atoms diffusing into Fe bulk are 0.03%, 0.035%, 0.05%, 0.08%, and 0.1%. Therefore, the influence of time on the diffusion distance is greater than that of temperature.

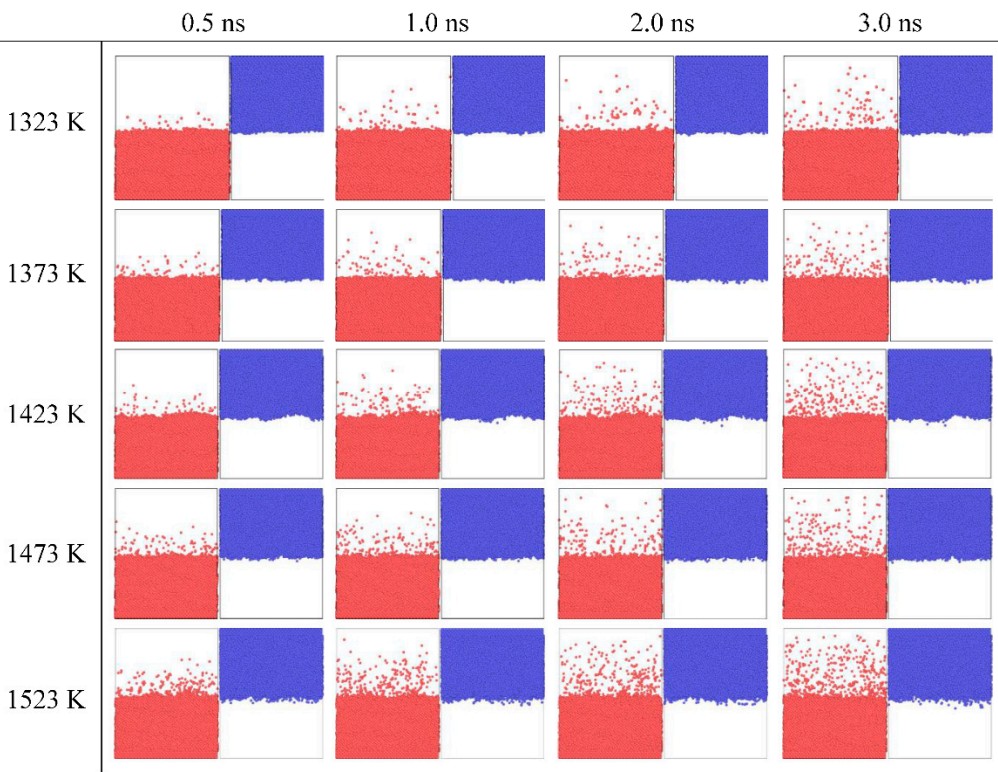

**Figure 2.** Snapshots of Fe atoms and Cu atoms under different diffusion times and temperatures. Red is Fe, blue is Cu.

Figure 3 illustrates the snapshots of the planar views of the equilibrium diffusion Fe/Cu interface at different temperature after 3 ns, respectively. As shown in Figure 3, when the temperature is increased, the Cu bulk presents an obvious liquid amorphous structure due to the disordered arrangement of atoms, while the Fe bulk maintains a solid crystalline structure. Furthermore, after the Cu atoms diffuse into the Fe lattice, the Cu atoms show an ordered state and occupy the lattice position of the Fe bulk, while the majority of the Fe atoms are located in the vacancies of the Cu lattice after diffusing into the inner Cu lattice. Thus, the Cu atoms gradually form agglomerations due to the increasing number of Fe atoms occupying more and more vacant positions in Cu bulk, explaining why the diffusion distance of Fe atoms diffusing into the Cu bulk is deeper than that of the Cu atoms diffusing into the Fe bulk.

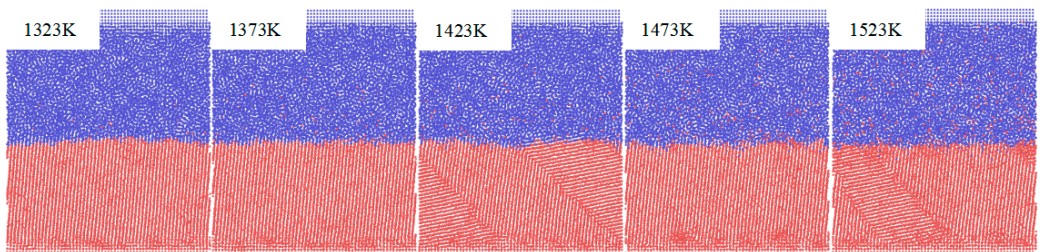

**Figure 3.** Atomic structure of Fe/Cu interface at different temperatures.

To deeply understand the non-uniform diffusion phenomena of the three Fe/Cu interface models, the MSDs of the Fe surface and the Cu surface in different directions at 1523 K were determined. As shown in Figure 4, the MSD curves of the Cu atoms in all orientations increase linearly, while the MSD curves of Fe atoms in different orientations areas fluctuate upward, which is consistent with the liquid and solid diffusion characteristics of the Cu and Fe matrixes. For the MSD curves of Fe atoms and Cu atoms in different directions, the MSD value of the diffusion of Cu atoms along the y-axis is higher than that along the x-axis and z-axis, indicating that Cu atoms mainly diffuse along the y-axis. The diffusion of Fe atoms on Fe (100) and Fe (111) surfaces is mainly along the x-axis, while on Fe (110) surfaces, the diffusion of Fe atoms is mainly along the y-axis. It can be seen that the degree of diffusion of the Fe/Cu interface in the diffusion process along the perpendicular direction is lower than that along the parallel direction of the interface.

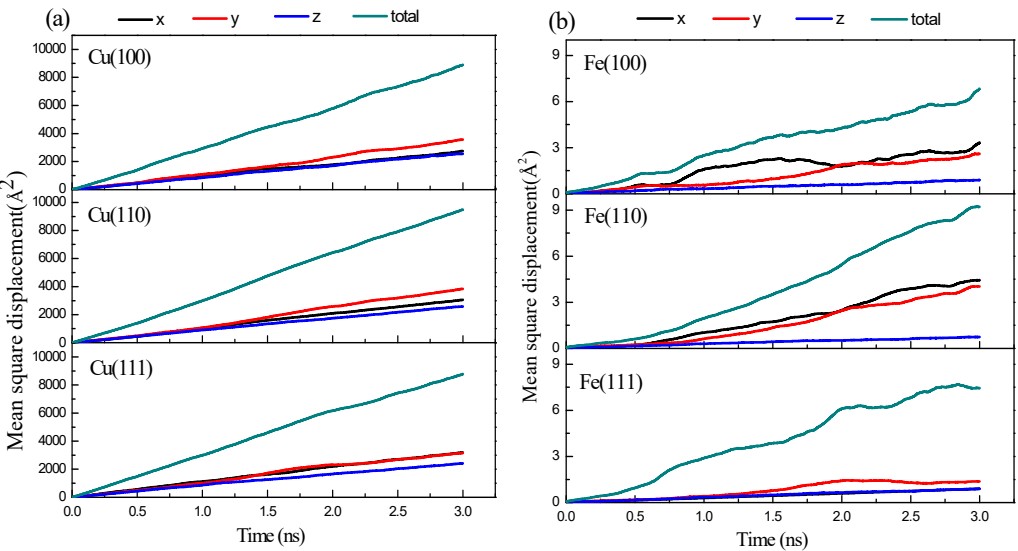

**Figure 4.** The mean square displacement curves of Fe/Cu interface at 1523 K with different orientation: (**a**) Cu (100), Cu (110), and Cu (111); (**b**) Fe (100), Fe (110), and Fe (100).

Table 1 shows the atomic diffusion coefficients of different surfaces and directions, and it can be seen that the diffusion coefficients of the Cu slabs in the z-axis direction can be ordered as (110) < (111) < (100), and the diffusion coefficients of Fe atoms in the z-axis direction can be ordered as (111) < (100) < (110). For all the Cu slabs, the diffusion coefficient of Cu atoms along y-axis is greater than that along the x-axis and z-axis, which can be ranked as follows: z < x < y, indicating that the blocking effect of Fe matrix on Cu atoms reduces the movement of Cu atoms along z-axis. However, for all the Fe slabs, the diffusion coefficient of Fe atoms along the z-axis is also smaller than other directions, which indicates that the transverse movement is greater than the axial movement during the diffusion process of Fe/Cu interface, and the Fe (110)/Cu (110) interface displays the best diffusion effect for the Fe atoms diffusing into the Cu matrix.

**Table 1.** Diffusion coefficient of atoms under different surface and orientation.

| Diffusion Direction | Cu (100) | Cu (110) | Cu (111) | Fe (100) | Fe (110) | Fe (111) |
|---|---|---|---|---|---|---|
| $D_x$ ($10^{-11}$ m²/s) | 409.5 | 523.6 | 528.6 | 0.803 | 0.296 | 0.745 |
| $D_y$ ($10^{-11}$ m²/s) | 605.7 | 735.2 | 626.6 | 0.456 | 0.575 | 0.813 |
| $D_z$ ($10^{-11}$ m²/s) | 431.1 | 421.8 | 389.3 | 0.126 | 0.159 | 0.117 |
| $D_{total}$ ($10^{-11}$ m²/s) | 491.7 | 549.6 | 512.6 | 0.353 | 0.464 | 0.554 |

In the in-depth investigation of the specific diffusion information of the Fe (110)/Cu (110) interface, the mean square displacement (MSD) of Cu atoms and Fe atoms along z-direction were calculated as shown in Figure 5. Additionally, the diffusion coefficient and the concentration distributions along the z-direction of Fe and Cu atoms was calculated to characterize the atomic diffusion at the Fe (100)/Cu (100) interface with temperature increase, as shown in Figure 5. One can see that all the MSD curves of the Cu atoms and Fe atoms are not only increased with increasing diffusion time at different temperatures, but also increased with increasing temperature due to the higher kinetic energy of atomic diffusion. Additionally, as shown in Figure 5a, the MSD curves of the Cu atoms present as straightly increasing trends as the diffusion time is incremented, but that of the Fe atoms does not maintain a straight line, as shown in Figure 5b. This is possibly due to the fact that the solid Fe matrix exhibits obvious volume instability when the ambient temperature approaches the melting point. From Figure 5c, it can be seen that the diffusion coefficient of Fe and Cu atoms exhibits a significantly increasing trend with increasing temperature, this is possibly due to an increase in the internal energy of the system. The diffusion coefficient of Cu atoms is far smaller than that of Fe atoms, seen in Figure 5c, but the diffusion depth of Fe atoms the diffuse into the Cu area is much extensive than that of Cu atoms, as seen in Figure 5d. This phenomenon concurs with the above results (see the snapshots in Figure 1), and is possibly due to the lower melting point of Cu than that of Fe. In contrast, the interface migrates to the Fe sides during the diffusion process. As shown in Figure 5d, the moving distance of the red dotted line is greater than that of the black dotted line. Thus, the interface migration may appear in these Fe/Cu interface models.

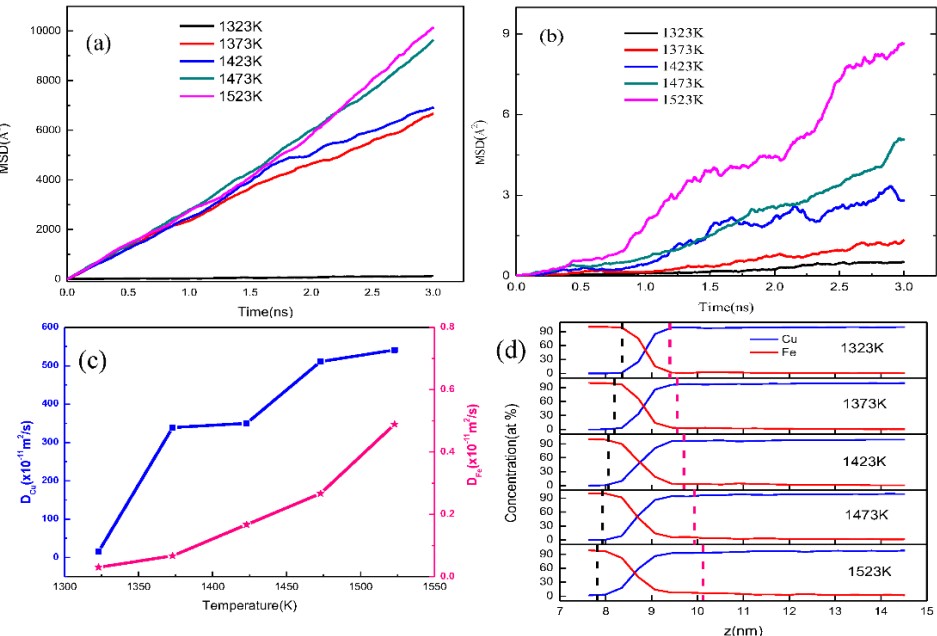

**Figure 5.** The diffusion behavior of atoms and the diffusion distance of Fe (110)/Cu (110) interface under different temperatures. (**a**) MSD curves of Cu atoms under different temperatures; (**b**) MSD curves of Fe atoms under different temperatures; (**c**) diffusion coefficient of Fe and Cu atoms; (**d**) atomic concentration of Fe and Cu atoms.

To clearly observe the phenomenon of interface migration, we compared the snapshots of the diffusion Fe/Cu interface at the initial simulation time and the final simulation time, as shown in Figure 6. Compared with Figure 6a,b, the Cu atoms become disordered in comparison to the initial model. The Fe atoms keep the original crystal structure, yet only a few Cu atoms replaced the position of Fe atoms, which elucidates the results of the concentration distributions. Furthermore, the red dashed line in Figure 6 migrates to the Fe side after the diffusion of Fe/Cu bimetallic material, indicating that Cu atoms are difficult to diffuse into the Fe matrix, regardless of displacement diffusion and gap diffusion. Hence, it has been demonstrated that the interface migration is authentically consistent in these Fe/Cu diffusion interface systems.

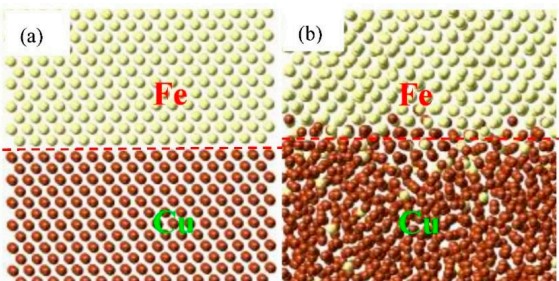

**Figure 6.** Atomic snapshots concerning Kirkendall effect of the Fe/Cu bimetallic interface. (**a**) Initial atomic structures; (**b**) the final atomic structures after the diffusion at 1523 K.

To deeply understand the inner diffusion mechanism of the Fe/Cu interface, the transition state search (TSS) tool was calculated to find the minimum energy and saddle points of the diffusion process. Figure 7 shows the diffusion energy barrier of five diffusion paths and the schematic diagram of the possible atomic diffusion paths near the interface boundary of the Fe and Cu atoms. Comparing five diffusion energy barriers in Figure 7, we can find that the diffusion of Cu and Fe atoms is mainly dominated by first-nearest-neighbor diffusion, and the diffusion activation energy of Fe atoms in the Cu matrix is greater than that of Cu atoms in the Fe matrix, indicating that the diffusion of Fe atoms in Cu matrix is easier than that of Cu atoms in Fe matrix.

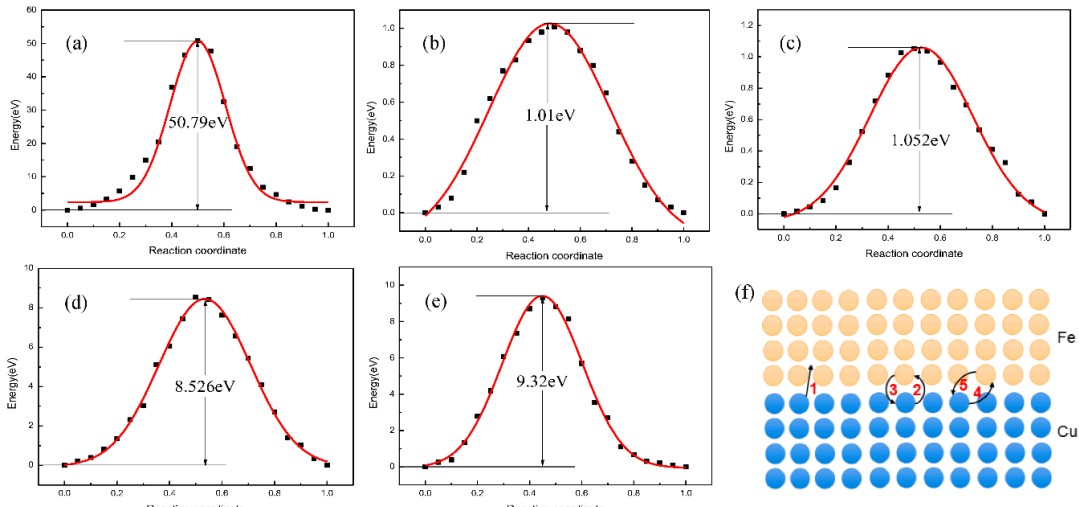

**Figure 7.** Atomic snapshots concerning Kirkendall effect of the Fe/Cu bimetallic interface. (**a**) the diffusion energy of the interstitial diffusion path of Cu atom; (**b**) the diffusion energy of the first nearest neighbor diffusion path of Cu atom; (**c**) the diffusion energy of the first nearest neighbor diffusion path of Fe atom; (**d**) the diffusion energy of the second nearest neighbor diffusion path of Cu atom; (**e**) the diffusion energy of the second nearest neighbor diffusion path of Cu atom; (**f**) schematic diagram of diffusion mechanism of Fe/Cu diffusion interface.

### 3.2. Mechanical Properties

3.2.1. Strain Rate and Orientation Effect

The strain rate's dependence of the stress–strain relationships of the Fe (100)/Cu (100) interface, the Fe (110)/Cu (110) interface, and the Fe (111)/Cu (111) interface under uniaxial tension are shown in Figure 8. As shown in Figure 8a, the strain–stress curves of the Fe (100)/Cu (100) interface are almost linear at the initial stage of uniaxial tension, which indicates that the Fe (100)/Cu (100) interface undergoes elastic deformation. After that, the stress decreases slowly with the increase in strain when the tension reaches ultimate strength, indicating that the Fe (100)/Cu (100) interface presented a plastic deformation ability and ductile fracture characteristics. From Figure 8b,c, one can see that the strain–stress curves are almost increasing linearly at the early stage in all the strain rates, which indicates that the Young's modulus is less sensitive to the strain rate. However, with the application of increasing strain, the curves present apparent ups and downs at the end of elastic deformation, which indicates that fracture occurred. Additionally, all the strain–stress curves decrease suddenly at the stress peak point at different strain rates, and no obvious plastic deformation can be observed. These results imply that the brittle fracture of the Fe (100)/Cu (100) interface and the Fe (111)/Cu (111) interface occurred at all simulated strain rates. Meanwhile, the yield strength and fracture strain both increase with the increasing strain rates, and the yield strength of three orientations can be arranged in the following order: Fe (111)/Cu (111) > Fe (110)/Cu (110) > Fe (100)/Cu (100), which reaches a remarkable agreement with the diffusion behavior of the three models.

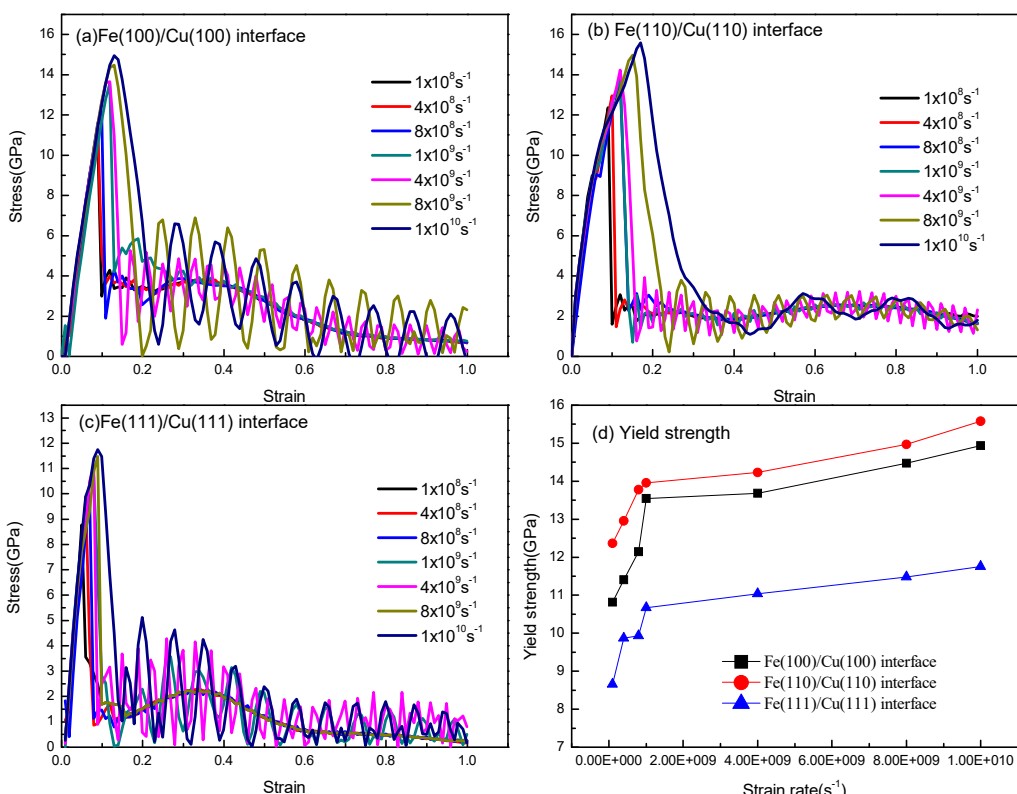

**Figure 8.** Stress–strain curves of (**a**) Fe (100)/Cu (100) interface at different strain rates from $1 \times 10^8$/s to $1 \times 10^{10}$/s; (**b**) Fe (110)/Cu (110) interface at different strain rates from $1 \times 10^8$/s to $1 \times 10^{10}$/s; (**c**) Fe (111)/Cu (111) interface at different strain rates from $1 \times 10^8$/s to $1 \times 10^{10}$/s; (**d**) the yield strength of three different interface models under different strain rates.

Referring to Figure 8a–c, the yield strength and fracture strain curves with different strain rates are described in Figure 8d. As shown in Figure 8d, the yield strength and fracture strain are particularly susceptible to strain rate when the strain stress is less than

$1 \times 10^9 \text{ s}^{-1}$. Nevertheless, when the strain rate exceeds $1 \times 10^9 \text{ s}^{-1}$, the tensile strength still increases slowly with the growth of strain rate, this is possibly due to the fact that the reinforcement effect is not statistically significant. Furthermore, the brittle fracture behavior of the three Fe/Cu interface models at a high strain rate is similar to that of single-crystal titanium [41].

All the atomic visualizations of the Fe/Cu interface are described using the Open Visualization Tool (OVITO), and all the inner structure of Fe/Cu bimetallic materials is characterized to observe the defect's behavior from HCP environments with common neighbor analysis (CNA) [42]. Additionally, the centrosymmetric parameter (CSP) and dislocation extraction algorithm (DXA) [43] analysis are powerful modules that were used to analyze the dislocation motion and defects of the Fe/Cu interface. To clearly describe the fracture mechanism of the Fe (100)/Cu (100) interface, the Fe (110)/Cu (110) interface and the Fe (111)/Cu (111) interface, the atomic snapshots (rendered by CNA, CSP, and DXA results) of the three interface models under uniaxial loading at different strain are described and shown in Figures 9–11.

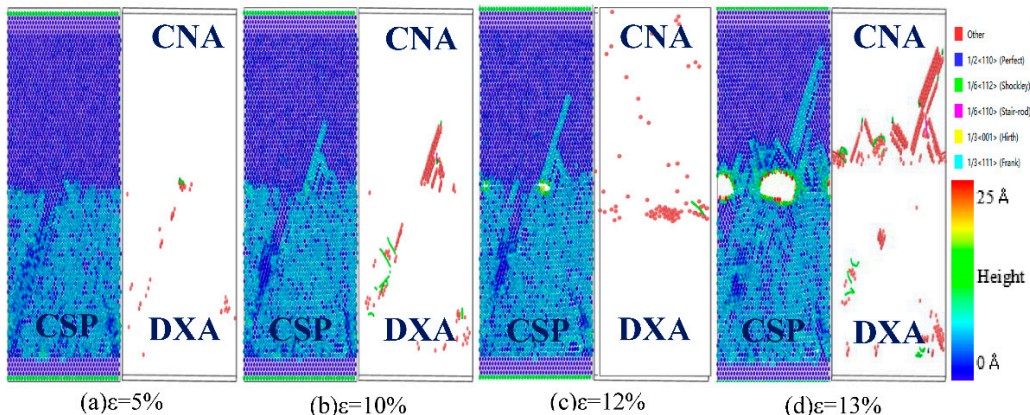

**Figure 9.** Atomic CSP, CNA, and DXA analysis of the Fe (100)/Cu (100) undiffused interface during z-axis tensile process.

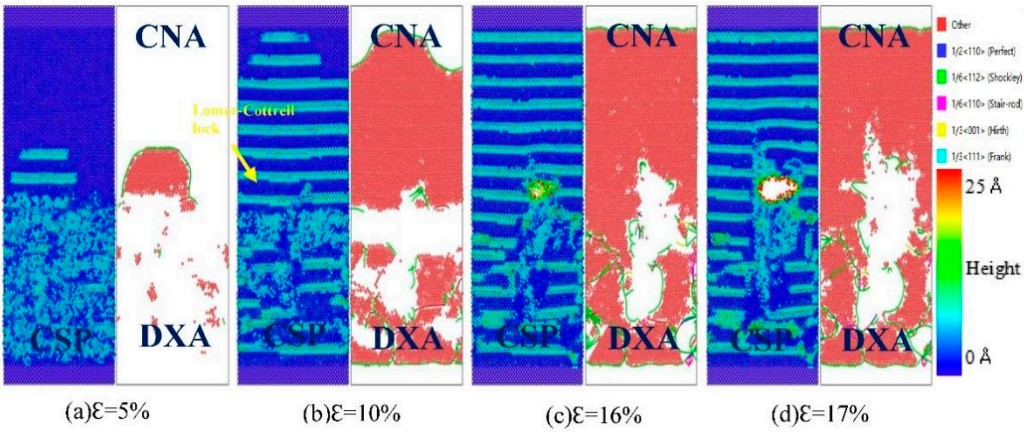

**Figure 10.** Atomic CSP, can, and DXA analysis of the Fe (110)/Cu (110) undiffused interface during z-axis tensile process.

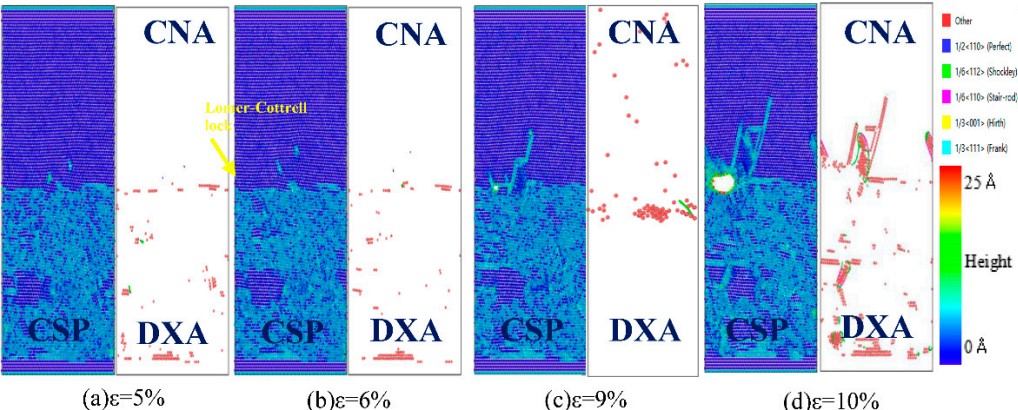

**Figure 11.** Atomic CSP, can, and DXA analysis of the Fe (111)/Cu (111) undiffused interface during z-axis tensile process.

From the CSP analysis in Figure 9, the atoms of the Fe (100)/Cu (100) interface are arranged neatly and represent an elastic deformation process on both sides of the interface at the beginning of the tension, and the interface structure hardly changes when the strain is lower than 5%. With gradually increasing stress, the Fe (100)/Cu (100) interface emits surface angular dislocations to the Cu side when the strain reaches 10%. Then, micropores are formed at the surface's angular dislocations when the strain reaches 12%. With continuous increase in strain, the interface gradually expands into a microcrack along the micropore, which results in a sudden drop of the interface stress value, and brittle fracture occurs. In addition, according to the CNA and DXA analysis, the stacking faults on the Cu side increase gradually with the progression of the Fe (100)/Cu (100) interface tension, and the evolution of the 1/6 <112> Shockley dislocations occupied a dominant position during the dislocation movement in the tensile process. Additionally, the number of Shockley dislocations near the interface increases sharply when a large number of cracks appear at the interface, while the number of Shockley dislocations is less before the fracture, indicating that brittle fracture begins at the strain of 13% in the Fe (100)/Cu (100) interface.

From Figure 10, it can be seen that a large number of stacking faults appear on both sides of the interface during the tensile fracture of Fe (110)/Cu (110) interface. Firstly, according to the CSP analysis, the steady stacking fault forms on the Cu side when the strain is 5%, but a large number of stacking faults appear on both sides of the Fe/Cu interface and surface angle dislocations are emitted to the Cu side when the strain increases to 10%. These results cause the development of micropores and cracks, resulting in interface fracture. Similarly, through integrated DXA analysis and CNA analysis, we can observe that a large number of stacking faults appear in the tensile process of Fe (110)/Cu (110) interface, and more 1/6 <112> Shockley dislocations appear near the interface. These phenomena hinder the progress of z-axis loading tensile, but improve the maximum stress value of the Fe (110)/Cu (110) interface.

In Figure 11, it can be seen that the Fe (111)/Cu (111) interface starts to emit Lomer–Cottrell lock to the Cu side when the strain reaches 6%, expands into micropores when the strain is 9%, and then expands into a microcrack when the strain is 10%. Comparing the tensile process with the two above interface models, one can see that there are fewer interlayer faults and Shockley dislocations at the Fe (111)/Cu (111) interface in the tensile process, resulting in lower yield strength and lower interface bonding strength of the Fe (111)/Cu (111) interface. To summarize, the yield strengths of the three orientation's interface models can be ranked as follows: Fe (110)/Cu (110) > Fe (100)/Cu (100) > Fe (111)/Cu (111). In addition, the Fe/Cu bimetallic interface shows brittle fracture characteristics in the tensile process, and the Fe (110)/Cu (110) interface shows a certain fracture toughness compared with the other two interface models.

### 3.2.2. Temperature Effect

To systematically study the influence of z-axis tensile temperature on the mechanical properties of Fe/Cu interface, the diffused Fe (110)/Cu (110) interface (the diffusion temperature is 1523 K, and the time is 3 ns) and undiffused Fe (110)/Cu (110) interface models were selected to carry out the z-axis tensile test at different temperatures (50, 100, 200, 300, 400, and 500 K). The stress–strain tensile results are shown in Figure 12; all these strain–stress curves show a linear and sharp increasing trend when the strain value is less than 10%, consistent with the characteristics of elastic deformation. Additionally, the slope of the stress–strain curve decreases with the increase in temperature, which indicates that the elastic modulus of the Fe/Cu interface decreases with the increase in temperature. Moreover, the Fe/Cu interface tension enters the plastic deformation stage after the yield strength, and the dislocation and the slip phenomenon are more prominent, and the tensile stress drops sharply and then changes slowly, indicating that brittle fracture occurs in the Fe (110)/Cu (110) interface before and after diffusion at all temperatures. Additionally, the maximum yield stress of the undiffused Fe (110)/Cu (110) interface at 50 and 500 K are 15.8 and 13.9 GPA, respectively, and the corresponding strain values are 18.9% and 16.1%, respectively. However, for the diffused Fe (110)/Cu (110) interface, the maximum yield stress values are 13.4 and 10.4 GPa, respectively, and the corresponding strain values are 11.9% and 10.7%, respectively. Moreover, the yield strength of the Fe (110)/Cu (110) interface is lower after interfacial diffusion, which is caused by the weakening of the interatomic bonding energy caused by the bimetal interface reconstruction after diffusion. When temperature increases from 50 to 100 K, the yield strength of the Fe (110)/Cu (110) interface decreases from 13.5 to 13.3 GPa respectively. However, when the temperature is raised from 400 to 500 K, the yield strength decreases from 11.2 to 10.3 GPa. Thus, the decreasing rates were 1.5% and 8.0%, respectively, indicating that the tensile Fe (110)/Cu (110) interface is more sensitive to temperature changes.

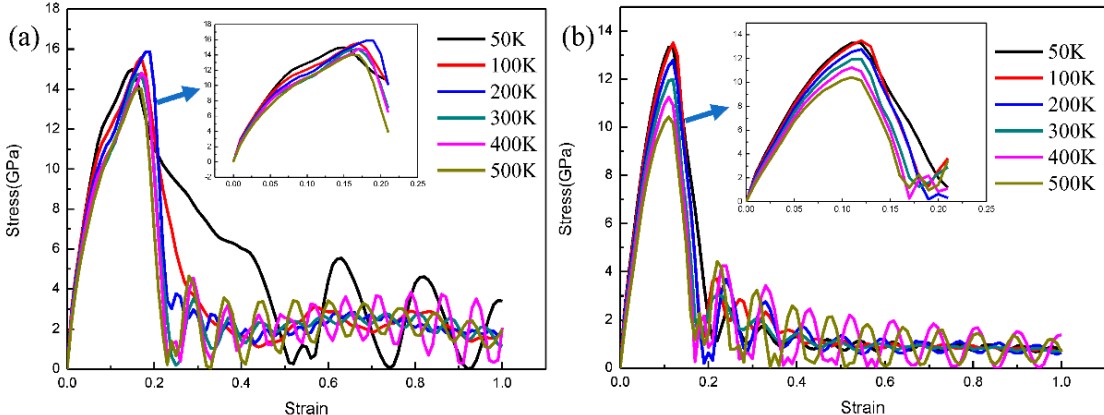

**Figure 12.** Strain–stress curves of z-axis tensile of Fe (111)/Cu (111) interface before and after diffusion under different temperature. (**a**) z-axis tensile stress–strain curve before diffusion; (**b**) z-axis tensile stress–strain curve after diffusion.

Thus, the atomic oscillation amplitude of Fe/Cu bimetallic interface is small when the tensile temperature is 50 K, resulting in a stable structure and a higher bonding strength. However, due to the increase in movement speed of atoms at the Fe/Cu bimetallic interface and the interaction force between atoms decreases when the tensile temperature is 500 K, a reduction in the energy required for dislocation, nucleation and emission, and lattice deformation occurred easily under the same load.

To reveal the fracture mechanism of the Fe (110)/Cu (110) interface at different working temperatures, the undiffused Fe (110)/Cu (110) interface at 300 and 500 K were selected to analyze the changes and atomic defects of the interface structure, and the results are shown in Figures 10 and 13 respectively. By comparing Figures 10 and 13, it can be found

that a large number of stacking faults also appear at the undiffused Fe (110)/Cu (110) interface at 500 K, which indicates that the main factor affecting stacking faults is the orientation relationship instead of tensile temperature. Additionally, the undiffused Fe (110)/Cu (110) interface emits a plane angular dislocation at the Cu side, and then evolves into a micropore when the strain is 10% and the stacking faults in the interface tensile process gradually decrease with the increase in temperature. This is possibly due to the higher tensile temperature, resulting in disordered structure arrangement and interface amorphization [44], which hinder the extended motion of dislocation and reduce the yield strength of the interface model. Although the number and length of dislocations increase or decrease, the types of the dislocations are all Shockley dislocations. Moreover, the fracture strain of the undiffused Fe (110)/Cu (110) interface decreases gradually with the gradual increase in temperature, possibly due to the early occurrence of interface fracture with the gradual increase in stress and the obstruction of dislocation propagation.

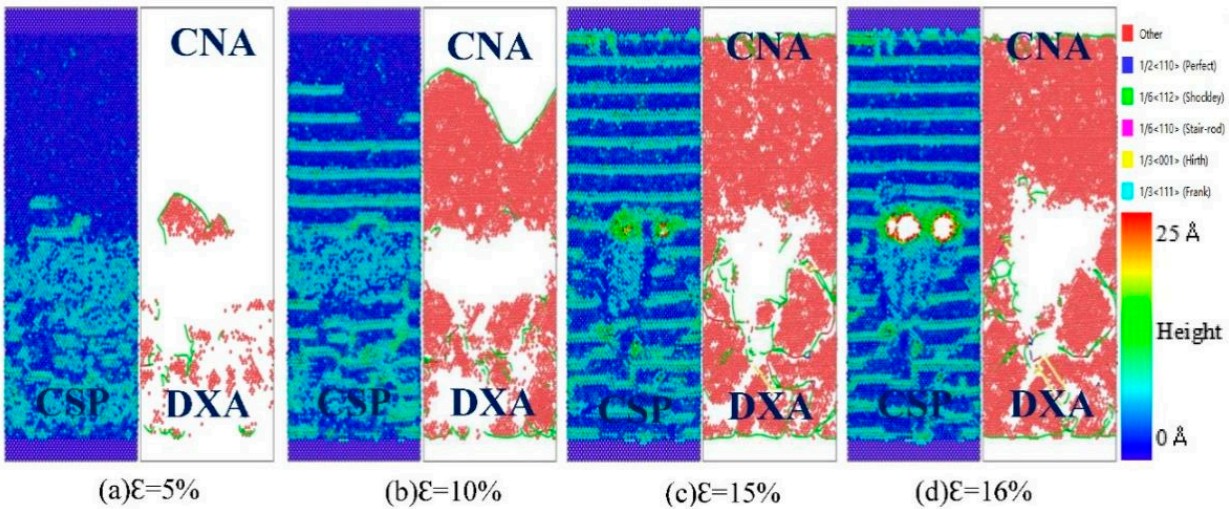

**Figure 13.** Atomic CSP, CNA, and DXA analysis of the Fe (110)/Cu (110) undiffused interface under z-axis tensile process at 500 K.

Figures 14 and 15 show the interface structure and atomic defect analysis during the tensile process of the diffused Fe (110)/Cu (110) interface at 300 and 500 K, respectively. From Figure 14, one can see that the micropores and microcrack at the diffused Fe (110)/Cu (110) interface appeared far from the interface on the Cu side, indicating that the bonding strength of the diffused Fe (110)/Cu (110) interface is greater than that of the Cu matrix. Similarly, the amorphization near the interface increase with increase in temperature, resulting in the reduction of interface stacking faults and the yield strength. From Figure 15, compared with the tension at 300 K, the fracture strain of the diffused Fe (110)/Cu (110) interface decreases from 20% to 19%, but the dislocation types are mainly 1/6 <112> Shockley dislocation at both 300 and 500 K. In summary, temperature will not change the dislocation and defect types of the Fe (110)/Cu (110) interface model under a certain temperature range, but will reduce or improve the yield strength and fracture strain of bimetallic interface by hindering or promoting the growth of dislocation.

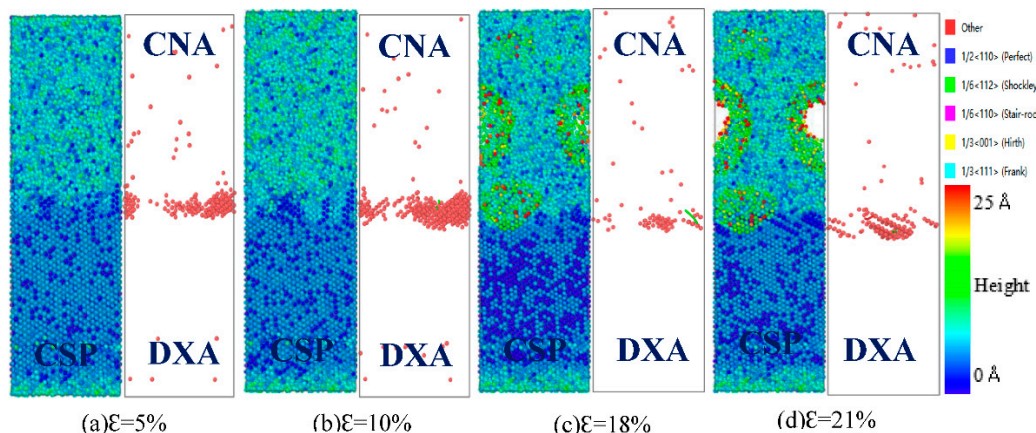

**Figure 14.** Atomic CSP, CNA, and DXA analysis of the Fe (110)/Cu (110) diffused interface under z-axis tensile process at 300 K.

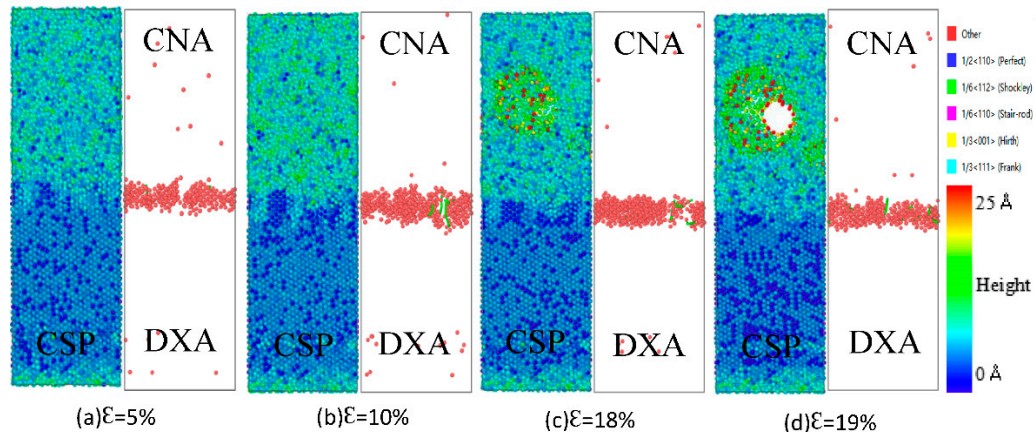

**Figure 15.** Atomic CSP, CNA, and DXA analysis of the Fe (110)/Cu (110) diffused interface under z-axis tensile process at 500 K.

## 4. Conclusions

In this research, the diffusion behavior, mechanical properties, and fracture mechanism of Fe/Cu solid–liquid interfaces were calculated by using a molecular dynamics simulation. The main conclusions obtained are as follows:

(1) Regarding the diffusion phenomenon of the Fe/Cu interface, it was observed that the diffusion distance increases with the increase in diffusion temperature and diffusion time. In addition, the diffusion distance of the Cu atoms diffusing into the Fe matrix is obviously less than that of the Fe atoms diffusing into the Cu matrix. The diffusion coefficient and the diffusion distance reach their maximums when the solid–liquid temperature is 1523 K, and the diffusion time is 3 ns.

(2) The diffusion coefficients of the Fe atoms when the temperature and time is 1523 K and 3 ns, respectively, are arranged in the following order: Fe (100) < Fe (110) < Fe (111). The diffusion coefficients of the Cu atoms are arranged in the following order: Cu (110) > Cu (111) > Cu (100).

(3) The yield strength and fracture strain of Fe/Cu bimetallic interface increase with the increase in strain rate and gradually decrease with the increase in tensile temperature. The yield strength of the three orientations can be ranked in the following order: Fe (110)/Cu (110) > Fe (100)/Cu (100) > Fe (111)/Cu (111). The Fe/Cu bimetallic interface shows brittle fracture characteristics during the tensile process, and the Fe (110)/Cu (110) interface shows a certain fracture toughness compared with the other two interface models.

(4) The yield strength of undiffused Fe/Cu bimetallic interface is higher than that of the diffused Fe/Cu interface. The number and length of 1/6<112> Shockley dislocations in the tensile process of the Fe/Cu bimetallic interface decreased after diffusion and the bonding performance of the Fe/Cu bimetallic interface is also reduced. In the undiffused Fe (110)/Cu (110) interface, bimetallic interface fractures occur along the interface, while in the diffused Fe (110)/Cu (110) interface, these occur on the Cu side, away from the interface.

**Author Contributions:** H.Z.: conceptualization, methodology, software; J.S.: visualization, investigation and supervision; N.G.: investigation; M.W.: investigation, writing, reviewing, and editing. All authors have read and agreed to the published version of the manuscript.

**Funding:** This research was funded by the Natural Science Foundation of Henan Province, China, under Grant No.: 212300410205.

**Institutional Review Board Statement:** Not applicable.

**Informed Consent Statement:** Not applicable.

**Data Availability Statement:** Some or all data, models, or code generated or used during the study are proprietary or confidential in nature and may only be provided with restrictions.

**Conflicts of Interest:** The authors declare no conflict of interest. The founders had no role in the design of the study; in the collection, analyses, or interpretation of the data; in the writing of the manuscript, or in the decision to publish the results.

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
