# Peer review of "Atomic Research on the Diffusion Behavior, Mechanical Properties and Fracture Mechanism of Fe/Cu Solid–Liquid Interface"

_coatings, doi:10.3390/coatings12091299_

Round 1

Reviewer 1 Report

I recommend the publication of the manuscript after a minor revision.

* Correspondence: - there is no name, no e-mail, and no co-author! 

Write keywords in alphabetical order.

Line 75: if possible, can you specify the mathematical model applied in this study?

Line 184: insert references for all mathematical formulas and explain all parameters.

Lines 347-351, 366-370, insert references and more details.

Specify the limits of this study. State the respective advantages and disadvantages.

The writing can be improved, it is still poor with numerous typos or grammar mistakes.

Insert Funding.

Competing Interests information: specify the grant codes. Re-write this paragraph.

Insert in References some articles that fit in this field published in “Coatings” journal, MDPI.

References are not written according to the Guide of Authors (ref. [4], [6], [10], [26], and so on).

If possible, I recommend the following references: 

[1] https://doi.org/10.3390/app12030946;

[2] https://doi.org/10.3390/app12094437.

This paper presents an interesting approach and deserved to be published after the mentioned revisions.

Reviewer 2 Report

Ref.comments to the paper titled as “Atomic research on the diffusion behavior, mechanical properties and fracture mechanism of Fe/Cu solid-liquid interface” written by the authors: Hongyu Zheng, Jingwen Sun, Na Guo and Mingjie Wang.

It should be remarked that the study of the different materials, including their interface is useful in order to extend our knowledge in the physical-chemical area. From this point of view the paper is modern and actual.

Introduction part. For the first, the authors have made the literature search, which contains 40 references. Some papers published last 5 years have been analyzed as well. Good!

Simulation methodology part. This LAMMPS method is classical, applied by the different scientific teams. It should be noticed that this approach has been used often and it has been applied with good advantage in the current paper as well. Atomic configuration of Fe/Cu interface model and the temperature-time relationship of the diffusion process, shown in Fig.1, is visualized the application of this method for this structure.

Results and Discussion part. Interesting results are shown in Table 1: Diffusion coefficient of atoms under different surface and orientation. Indeed, the estimation of the diffusion coefficient for the different orientation of the Fe and Cu atoms can be useful in the practical area for the engineers. About the data presented in Figure 5: The diffusion behavior of atoms and diffusion distance of Fe (110)/Cu (110) interface under different temperatures – I would like to ask the authors about the following. Have you’ an additional data on the manifestation of the drift mechanism? It seems to me and according to my experience, after the change of the temperature at this interface the field gradient can be appeared, thus you can registered a little value of the voltage, that influence on the manifestation of the drift mechanism. Please explain the lack of the drift mechanism manifestation.

Mechanical properties testing part. Figure 8: Stress-strain curves …Very nice results shown namely in part d of this Figure.

Temperature effect part. Please see my question below about the possible drift mechanism including.

Conclusion part accumulated the main results.

Please check the Ref.list. All authors mane should be written with a capital letter (please, for example, ref.[40].

As for my local opinion the paper should be revised (minor) according the questions before the publication.
